# The Prevalence of Cancer Predisposition Syndromes (CPSs) in Children with a Neoplasm: A Cohort Study in a Central and Eastern European Population

**DOI:** 10.3390/genes15091141

**Published:** 2024-08-29

**Authors:** Gabriela Telman-Kołodziejczyk, Ewa Strauss, Patrycja Sosnowska-Sienkiewicz, Danuta Januszkiewicz-Lewandowska

**Affiliations:** 1Department of Pediatric Oncology, Hematology and Transplantology, Poznan University of Medical Sciences, Szpitalna 27/33, 60-572 Poznan, Poland; gtelman@skp.ump.edu.pl; 2Institute of Human Genetics, Polish Academy of Sciences, Strzeszynska Street 32, 60-479 Poznan, Poland; strauss@man.poznan.pl; 3Department of Pediatric Surgery, Traumatology and Urology, Poznan University of Medical Sciences, Szpitalna Street 27/33, 60-572 Poznan, Poland; sosnowska@ump.edu.pl

**Keywords:** cancer, children, cancer predisposition syndromes (CPSs)

## Abstract

Importance: The etiology of pediatric cancers is often unclear; however, advancements in genetics have identified significant roles for genetic disorders in their development. Over time, the number of cancer predisposition syndromes (CPSs) and awareness of them have increased, providing the possibility of cancer prevention and early detection. Purpose: In this study, we present data concerning the number and type of oncological cases and their correlation with CPS occurrence in a cohort of Central and Eastern European pediatric patients. Materials: The data were collected between 2000 and 2019 at the Karol Jonscher Clinical Hospital of Poznan University of Medical Sciences, resulting in a cohort of 2190 cases in total, of which 193 children (8.81%) were confirmed to have a CPS. Results: CPSs occurred most frequently in infancy (22.90% of all children suffering from any diagnosed cancer during the first year of life; *p* < 0.0001), accounting for more than one-quarter of all CPS cases in our cohort. CPSs were least likely to be observed in patients aged 14 and 15 years (2.17% and 2.44% of children diagnosed with any of the listed cancers at the exact age, respectively; *p* < 0.05). Among CPSs, the most common were neurofibromatosis type I (NF1), Li–Fraumeni syndrome (LFS), and Down syndrome (DS). Conclusions: To conclude, it is important to emphasize the need for personalized treatment for each patient affected by both CPSs and subsequent cancer in order to reduce the toxicity of therapy and improve quality of life by reducing the risk of side effects.

## 1. Introduction

Pediatric cancers are relatively rare, with an estimated annual incidence of up to 400,000 cases globally. Cancers in children remain rare diseases, even though the researchers’ model estimated there being up to 400,000 newly diagnosed childhood neoplasms worldwide annually [1]. The exact cause of a cancer diagnosis very often stays unknown, making it difficult to prevent in pediatric patients. In the early 1970s, Knudson proposed the two-hit hypothesis, which involves the inactivation of both alleles of a tumor suppressor gene, leading to cancer development (by mutation or epigenetic silencing) as causatives of phenotype changes [2,3,4]. Thanks to medical and technological development, the role of genetics in neoplasm pathogenesis is becoming more and more crucial, especially regarding cancer predisposition syndromes (CPSs), whose number and consciousness increase with passing time. Depending on the data source, the incidence of CPSs is between 7 and 18% in the pediatric population suffering from neoplasms in general, with up to 21% of these children having central nervous system (CNS) tumors [5,6,7,8,9,10,11,12,13]. Currently, genetic counseling and testing should be one of the routine actions during neoplasm diagnosis in the pediatric population, at least if there are prerequisites of an underlying CPS, such as young age of cancer diagnosis, phenotype features, atypical neoplasm diagnosis, positive family history, or being a cancer survivor. Although several research groups have tested previously created questionnaires to facilitate the identification of relevant cancer mutations, none of these questionnaires have achieved 100% sensitivity. Unfortunately, this lack of sensitivity could contribute to missed diagnoses of CPSs [9,14,15,16,17] as well as less accessibility and higher costs for genetic testing in previous decades, when the first cohort analyses were carried out, usually with shorter follow-ups.

One of the most common and known CPSs is Down syndrome (DS). Many recommendations and therapy modifications have been made according to an underlying genetic disorder, which has benefited affected children and their families in early cancer detection, reducing toxicity during treatment and, in the end, improving survival [18,19]. These seem to be the available advantages of thoroughly understanding any CPS only if the proper effort is made in many medical centers, especially because some CPSs are extremely rare but still worth detecting and understanding better.

In this study, we present analyzed data concerning the number and type of oncological cases and their correlation with CPS occurrence among a cohort in a Central and Eastern European pediatric population. To date, this is the first paper summarizing 20 years of data collection in a uniform cohort of children and adolescents and, as a result, highlighting the need for awareness of CPSs as a probable cause of related neoplasms. Since pediatricians and general practitioners are often the first physicians to encounter a child suspected of having an underlying molecular disorder (such as a CPS), they have deep insight into a family’s history and play an essential role in vigilant follow-ups for the prevention of serious diseases. We believe that they will find it crucial to detect and refer such patients for genetic counseling. Also, through our work, we aim to encourage oncologists to consider genetic testing more frequently, recognizing that the results can benefit patients and their families by enabling therapy modifications, reducing side effects, providing personalized recommendations, and ensuring vigilant cancer surveillance screening for second primary neoplasms. This is particularly important, as CPS-positive children and adolescents are at high risk of developing such neoplasms [20,21,22,23,24].

## 2. Materials and Methods

### 2.1. Study Population

This study represents 2190 unselected Caucasian children from birth up to the 18th year of age who were hospitalized with newly diagnosed neoplasms at Karol Jonscher’s Clinical Hospital of the Poznan University of Medical Sciences between 2000 and 2019. Giving that diagnosis on the day of one’s 18th birthday or any time after in Poland would lead to the patient being taken under adult medical care in another hospital, we do not disclose any neoplasm incidence information about patients who were 18 years old or older. Due to analyzing retrospective latent cancer incidence data, signed permission for including the patients in our cohort study was not required from either the legal guardians or the affected children. This study was approved by the Bioethics Committee of the Medical University of Poznan (Resolution No. KB-549/24).

### 2.2. Clinical Evaluation

The study cohort inclusion criteria required confirmed neoplasms. A cancer diagnosis led to the assigning of appropriate ICD-10 codes from C00 to D48, and it was the basis for enrolling a patient in our cohort, which at first counted 3663 patients. After detecting and removing (1) replicated cases, (2) duplicate cases, (3) patients previously diagnosed in other regions who were transferred to our hospital to continue treatment initiated elsewhere (e.g., to undergo hematopoietic stem cell transplant), and (4) children diagnosed before 1 January 2000, the final cohort consisted of 2190 individuals (1233 males and 957 females).

The diagnosis of cancer predisposition syndromes (CPSs) was based on (1) a phenotypic trait, such as those observed in Down syndrome (DS), neurofibromatosis type 1 (NF1), or isolated hemihypertrophy (IHH), often confirmed by genetic disorder testing, and (2) genetic testing, including molecular analyses targeting mutations within a single gene, Sanger Sequencing, Whole Exome Sequencing (WES), and Next-generation Sequencing (NGS) using the TriSight One Expanded Sequencing Panel by Illumina. This panel allows for the analysis of 6699 genes with confirmed clinical significance, from which the oncology gene subpanel (comprising 125 genes) was selected. Genetic testing was performed in Polish centers (Lodz, Poznan, Szczecin, and Warsaw), as well as abroad in Germany, Italy, Norway, the United Kingdom, and the USA.

### 2.3. Data Analysis

Categorical data were presented as numbers (n) and percentages (%). Univariate analyses were used to compare the distribution of studied parameters’ ꭓ^2^ test, with or without Yates’ correction, depending on the number of cases. Odds ratios (ORs) and 95% confidence intervals (95% CIs) were calculated to assess the association between the presence of CPS-related neoplasms. Based on the OR and statistical significance model, the best-fitting model was proposed. Statistical analyses and graphs were generated using GraphPad Prism v6 software. Differences were considered significant when *p* < 0.05. 

## 3. Results

### 3.1. The General Characteristics of Patients

Over a 20-year retrospective follow-up 2190 children were newly diagnosed with cancer in Karol Jonscher’s Clinical Hospital of Poznan University of Medical Sciences, a highly specialized pediatric hospital in the western region of Poland. Table 1 summarizes detailed characteristics of the children in this study cohort, including sex, age, and the total number of exact diagnoses according to ICD-10 classification. The largest group consisted of leukemia patients (codes C91 and C92), totaling 692 (31.60%). The second most common group was patients with central nervous system (CNS) tumors (codes C70 to C72), with a total of 382 (17.44%). The third largest group was lymphoma patients (codes C81 to C85), totaling 279 (12.74%). Together, these three groups accounted for over 61% of all newly diagnosed patients in our retrospective analysis.

### 3.2. Prevalence of Specific Cancers Overall and by Age Group

Table 2 presents the prevalence of common and rare neoplasms in children based on age, as defined by the ICD-10 classification. The distinguished cancers are (1) lymphoid leukemias (C91) and (2) myeloid leukemias (C92), with 505 (23.06%) and 187 (8.54%) patients, respectively; (3) central nervous system tumors (C70–C72) with a total of 382 (17.44%) children; (4) Hodgkin lymphomas (C81) and (5) non-follicular lymphomas (C83), with 268 (12.24%) cases overall, including 144 (6.58%) Hodgkin and 124 (5.66%) non-follicular lymphomas; (6) malignances of the peripheral nerves and autonomic nervous system, such as neuroblastomas (C47), affecting 177 (8.08%) children; (7) soft tissue tumors (C49), with rhabdomyosarcoma (RMS) being the most frequently diagnosed, totaling 161 (7.35%) patients; and (8) malignant neoplasm of the kidney (C64), with Wilms tumor being the most common, accounting for 130 children (5.94%). The diagnoses mentioned were given to 1810 patients, which accounts for over 82% of all children. The remaining 380 patients had less common neoplasms, including: cancer of the endocrine glands (C73–C75), with thyroid and adrenal tumors being the most frequent, totaling 95 cases (4.33%); bone and articular cartilage neoplasms (C40–C41), affecting 64 children (2.92%); leukemias not previously mentioned and other malignancies of lymphoid, hematopoietic, and related tissue (C94–C96), comprising 61 patients (2.79%); liver and intrahepatic bile duct cancers (C22), with hepatoblastoma being the most common, totaling 42 cases (1.92%); and gonadal neoplasms (ovary (C56) and testis (C62)), with a total of 44 cases (2.01%), including 21 females (0.96%) and 23 males (1.05%). Other cancers with a prevalence of less than 1% in our cohort are considered extremely rare in the pediatric population and are listed in Table 1 with an exact number of cases.

A higher prevalence of specific cancers was observed in different age groups, as demonstrated in Table 2. Tumors of the peripheral nerves and autonomic nervous system (C47), soft tissue tumors (C49), kidney cancers (C64), central nervous system tumors (C70–C72), and leukemias (both C91 and C92) were predominantly diagnosed during infancy and toddlerhood. Later, in early childhood, the frequency of neoplasms classified as C47, C49, C70–C72, and C92 began to decrease, while those classified as C64 and C91 increased, reaching their highest prevalence in our cohort. As mentioned earlier, the first peak of cancer incidence in the entire cohort occurred from birth through early childhood. While the frequency of neoplasms classified as C70–C72 and C91 (lymphoid leukemia) continued to decrease during pre-school and early school years, the first peak of non-follicular lymphoma (C83) incidence was observed. The second peak for non-follicular lymphomas was noted in the age range of 13 and 15 years. The prevalence of Hodgkin lymphoma (C81) and myeloid leukemia (C92) began to increase significantly during pre-puberty and early adolescence (up to 14 years old). This trend led to a peak in Hodgkin lymphoma (C81) incidence among adolescents, while the frequency of myeloid leukemia (C92) remained relatively constant up to the age of 17. Consequently, a second peak in overall cancer incidence was observed, including some rarely detected cancers in children that are more common in adults: bladder cancer (C67), thyroid gland tumor (C73), bone and articular cartilage cancer (C40–C41), and testis neoplasm (C62).

In summary, two peaks of cancer incidence were observed in our pediatric cohort: (1) in early childhood (from birth up to 5 years old), with a total of 1013 children (46.3%), and (2) in adolescence (between 15 and 17 years old), with a total of 370 adolescents (16.9%). These results are presented in the last column of Table 2, showing a prevalence range of 5.0% to 9.9% of the total number of patients at the specified age. In contrast, the prevalence of neoplasm in children of other ages did not exceed 4.7%.

### 3.3. Incidence of Cancer Predisposition Syndromes (CPSs) According to Age

Over the 20-year study follow-up, cancer predisposition syndromes (CPSs) were detected in 193 out of 2190 patients (8.81%). The data on the incidence of neoplasm and CPSs by age are presented in Table 3 and Figure 1.

Cancer predisposition syndromes (CPSs) occurred most frequently in infancy, with 22.90% of all children diagnosed with cancer during the first year of life having CPSs (*p* < 0.0001). This represents more than one-quarter of all CPS cases in our cohort. Within this group, CPSs were most commonly detected in 9-month-olds (41.67% of 12 cases diagnosed with neoplasm at this age, *p* < 0.001), followed by newborns up to 1 month old (25.00% with *p* = 0.001) and 5 months old (26.32% of 19 patients at this age, *p* < 0.05). CPSs continued to be relatively common during toddlerhood, occurring in 15.72% of 3-year-olds (*p* < 0.01) and 13.74% of 2-year-olds (*p* < 0.05) diagnosed with cancer. Conversely, CPSs were least likely to be observed in 14- and 15-year-olds, with prevalence rates of 2.17% and 2.44%, respectively, among children diagnosed with any of the previously listed neoplasms at these ages (*p* < 0.05). In other age groups, CPSs were diagnosed with intermediate frequency, though these results may not be statistically significant (*p* > 0.05).

The list of CPSs detected in our cohort, categorized by age, is presented in Appendix A. Among infants, neurofibromatosis type I (NF1) was the most frequently observed CPS, accounting for 23 cases (46.94% of CPS-positive patients in that age group), particularly within the first two months of life. During this period, NF1 was observed in 11 newborns and 1-month-olds (47.83% of all NF1 cases in infancy). Down syndrome (DS) was also relatively common in this age group, being present in all three cases detected during infancy. Other CPSs detected during the first year of life included isolated hemihypertrophy (IHH), with six cases, making it the most frequently diagnosed CPS among 9-month-olds; Beckwith–Wiedemann Syndrome (BWS); and Li–Fraumeni Syndrome (LFS), each with five cases, with BWS being more prevalent among 5-month-olds. In toddlerhood, between 2 and 3 years old, NF1 remained the most frequently diagnosed CPS, with 15 cases. LFS and DS were also observed, with ten and seven cases, respectively. Other CPSs detected during this period included BWS and IHH, each with five cases.

To be more specific, in 3-year-olds, constitutional mismatch repair deficiency syndrome (CMMRDS) was detected, which was not observed at any other age group, and hemophagocytic lymphohistiocytosis (HLH) was found in two cases. Among 14- and 15-year-olds, a group less likely to be diagnosed with CPS-related cancer, NF1 was detected in three cases overall, HLH in one 14-year-old, and multiple endocrine neoplasia syndrome (MEN) in one 15-year-old, totaling five CPS cases during these two years of life.

### 3.4. Observed Age Ranges for the Highest Risk of CPS-Related Cancer Development in Children

We divided our cohort into four age ranges for statistical analysis. The designated age limits were chosen as the model best suited to the study population. 

Children aged 11 to 18 years had the lowest risk of developing CPS-associated neoplasm (17% vs. 35% for non-CPS related cancers) (Table 4). In comparison, the highest risk for CPS-related cancers was observed in the following age groups: infants from 0 to 1 month old (7.52-fold higher risk), children from 2 to 12 months old (5.81-fold higher risk), and those between 1 and 11 years olds (2.08-fold higher risk). 

### 3.5. Neoplasms Diagnosed in CPS-Positive Patients

Appendix A includes data on diagnosed cancers among 193 CPS-positive patients, categorized according to ICD-10 classification. Among all detected cancer predisposition syndromes (CPSs), neurofibromatosis type I (NF1) was the most prevalent, with 83 patients. NF1 was notably common in patients with malignancies of peripheral nerves and autonomic nervous system (C47), central nervous system (CNS) tumors (C71–C72), and malignant neoplasms of other connective and soft tissues (C49). The second most common CPS was Li–Fraumeni Syndrome (LFS), with 34 cases. The largest group within this category had C71, C47, and malignant neoplasms of liver and intrahepatic bile ducts (C22). Additionally, two LFS-positive patients with multiple cancers were observed, both diagnosed with rhabdomyosarcoma and myeloid leukemia. Down syndrome, the third most notable CPS, was present in 19 cases, all affected by leukemias, including both lymphoid (C91) and myeloid (C92). Notably, all patients diagnosed with isolated hemihypertrophy (IHH) had confirmed malignant neoplasm of the kidney (C64). Beckwith–Wiedemann Syndrome (BWS) was identified in ten cases, with seven patients diagnosed with C64. Other less frequently observed CPSs are listed in Appendix A.

## 4. Discussion

To our knowledge, this is the first study summarizing the incidence of cancer predisposition syndromes (CPSs) based on childhood neoplasms in an unselected Central and Eastern European population, with 20 years of data collection. In our cohort of 2190 cancer-positive pediatric patients, 193 (8.81%) were diagnosed with an underlying cancer predisposition syndrome (CPS). Current estimates suggest that CPS prevalence ranged from 7% to 18% in the general pediatric population, in general, and could be as high as 21% among children with central nervous system (CNS) tumors [5,6,7,8,9,10,11,12,13]. We believe that CPSs have been present in our population over the past several decades at rates comparable to those observed today. However, the ability to detect them was limited. What is more, during those 20 years, the detection capabilities were historically limited. Additionally, over the 20 years of our study, the availability of genetic testing improved [25,26,27,28,29], but it remains underfunded in Poland. Consequently, it is often inaccessible to less affluent families or those dependent on public hospital funding. This limitation may contribute to an underestimation of PS prevalence. Furthermore, the incidence of cancer in our cohort may be influenced by medical tourism, both domestically and abroad, which can be affected by factors such as residence, financial resources, and other variables [30,31]. These factors represent limitations of the study that should be considered in future research.

As long as detailed genetic testing remains underfunded and specialist counseling is limited, coupled with the absence of standardized CPS screening algorithms across countries, different medical centers may conduct diagnostics with varying frequency. This variability can lead to a wide range of CPS prevalence estimates and unfortunately means that the total incidence of CPS is likely to be underestimated. Given the undeniable benefits of personalized treatment for CPS, it is crucial that the costs of all related diagnostic efforts be funded by the government. This is particularly important in the context of an aging global population [32] and the improving curability of cancers in children and adolescents.

As noted earlier, we observed two peaks in cancer incidence within the studied population, which aligns with findings from other researchers [33,34]. The first peak, occurring during the first five years of life, may be associated with the high prevalence of blastomas, which arise from malignance processes in precursor cells. This group includes neuroblastoma, Wilms tumor, medulloblastoma, hepatoblastoma, and retinoblastoma, which are common early childhood tumors, but become less common as children grow older. The second peak, observed during adolescence, corresponds with the biological maturation process, where individuals begin to resemble adults more than children. This period is marked by an increased incidence of lymphomas, myeloid leukemias, thyroid cancers, testis neoplasms, and bladder malignances, which are also more prevalent in the population [35,36]. 

It is worth noting that the number of childhood cancer survivors among adolescents and adults is increasing over time due to earlier detection and continuously improving treatment strategies [37,38]. As a result, more individuals may experience side effects related to their previous cancer diagnosis and the therapies used, with the frequency of these effects increasing as time passes from the initial diagnosis [39,40,41,42,43]. One of the most concerning late complications is the development of secondary malignancy, which poses a particularly high risk for patients with CPSs [20,21,22,23,24]. To effectively manage these health consequences and treatment responsibilities related to CPSs, patients need to receive genetic counseling and be diagnosed with any molecular disorder first.

In our cohort, diagnostic tests for CPSs were under specific conditions, such as the occurrence of rare and atypical cancers, distinctive phenotypic features, recurrence or multiplied neoplasm, and a significant family history of cancer. When these conditions were met, genetic testing, including molecular analysis targeting mutations within a single gene, Sanger Sequencing, Whole Exome Sequencing (WES), or Next-generation Sequencing (NGS), was offered. Genetic counseling was frequently conducted following the diagnosis of neoplasm. Early detection is crucial for effective preventive care, as it allows for tailoring treatment, particularly for neoplasms, to the underlying mutation. It is likely that CPS diagnosis should be considered more frequently than current algorithms suggest [9,14,15,16,17,44,45]. This is especially important given the advancements in genomic screening that aid in the detection of CPS-related neoplasm [12,18,21,29,46]. Both our research and that of other groups emphasize the need for personalized treatment for patients affected by both CPS and subsequent neoplasm. Personalized treatment can help reduce therapy toxicity and improve the quality of life by minimizing the risk of side effects [47,48,49]. We recommend adhering to the latest guidelines, which have been updated in recent years [19,50,51,52,53,54,55].

## 5. Conclusions

A higher prevalence of cancer predisposition syndromes (CPSs) in infancy and toddlerhood may be linked to inherited genetic disorders. According to Knudson’s two-hit theory, the development of neoplasm can occur right after the second mutation (i.e., second hit), with the first being associated with the presence of a CPS. While Knudson’s theory does not explain the molecular basis of every inherited neoplasm in children, it remains a fundamental framework for advances in genomic research [56]. Nevertheless, recognizing the possibility of CPS offers an opportunity for careful and personalized monitoring, which can lead to early cancer detection or even prevention by reducing carcinogenic environmental factors and implementing targeted treatment strategies [57]. This study, the first of its kind in the cohort of children and adolescents from Eastern and Central Europe, details the incidence of cancers and related CPSs in our region. Moreover, our 20-year retrospective study is among the few globally with such an extensive CPS data collection period. These results are consistent with previously documented CPS prevalence in children and adolescents [5,6,7,8,9,10,11,12,13]. These findings should prompt physicians to intensify screening for CPSs, as their incidence remains underestimated. We believe this could be a significant focus for further research.

## Figures and Tables

**Figure 1 genes-15-01141-f001:**
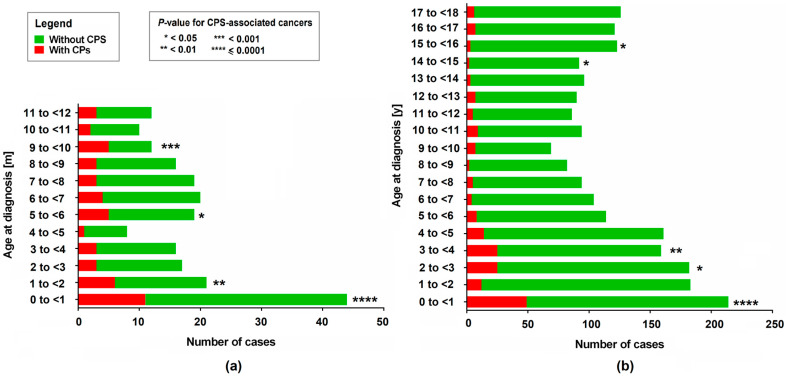
Identification of 193 cancer predisposition syndromes (CPSs) in a regional Caucasian unselected cohort of 2190 children with neoplasms from a Central and Eastern European population, both (**a**) in infants (from birth up to 1 year old) and (**b**) in the entire studied cohort.

**Table 1 genes-15-01141-t001:** The general characteristics of the studied patient cohort includes sex, age group distribution at the time of diagnosis, and the total number of specific diagnoses according to the ICD-10 classification.

Characteristic		No. (%)
Sex		
Male		1233 (56.30)
Female		957 (43.70)
Age at diagnosis of neoplasm, years old [y]		
0 to <5 y		899 (41.05)
5 to <10 y		463 (21.14)
10 to <15 y		458 (20.91)
15 to <18 y		370 (16.89)
Diagnosis		
ICD-10	Mayor diagnosis	
C11	Malignant neoplasm of nasopharynx	4 (0.18)
C22	Malignant neoplasm of liver and intrahepatic bile ducts	42 (1.92)
C30	Malignant neoplasm of the nasal cavity and middle ear	2 (0.09)
C37	Malignant neoplasm of thymus	1 (0.05)
C38	Malignant neoplasm of the heart, mediastinum and pleura	7 (0.32)
C39	Malignant neoplasm of other and ill-defined sites in the respiratory system and intrathoracic organs	1 (0.05)
C40	Malignant neoplasm of bone and articular cartilage of limbs	37 (1.69)
C41	Malignant neoplasm of bone and articular cartilage of other and unspecified sites	27 (1.23)
C43	Malignant melanoma of the skin	1 (0.05)
C47	Malignant neoplasm of peripheral nerves and autonomic nervous system	177 (8.08)
C48	Malignant neoplasm of retro-peritoneum and peritoneum	4 (0.18)
C49	Malignant neoplasm of other connective and soft tissue	161 (7.35)
C56	Malignant neoplasm of ovary	21 (0.96)
C62	Malignant neoplasm of testis	23 (1.05)
C64	Malignant neoplasm of kidney, except renal pelvis	130 (5.94)
C67	Malignant neoplasm of bladder	4 (0.18)
C68	Malignant neoplasm of other and unspecified urinary organs	1 (0.05)
C69	Malignant neoplasm of the eye and adnexa	12 (0.55)
C70	Malignant neoplasm of meninges	5 (0.23)
C71	Malignant neoplasm of the brain	357 (16.30)
C72	Malignant neoplasm of the spinal cord, cranial nerves and other parts of the central nervous system	20 (0.91)
C73	Malignant neoplasm of the thyroid gland	73 (3.33)
C74	Malignant neoplasm of the adrenal gland	17 (0.78)
C75	Malignant neoplasm of other endocrine glands and related structures	5 (0.23)
C76	Malignant neoplasm of other and ill-defined sites	13 (0.59)
C79	Secondary malignant neoplasm of other and unspecified sites	3 (0.14)
C80	Malignant neoplasm, without specification of the site	1 (0.05)
C81	Hodgkin lymphoma	144 (6.58)
C83	Non-follicular lymphoma	124 (5.66)
C84	Mature T/NK-cell lymphomas	1 (0.05)
C85	Other and unspecified types of non-Hodgkin lymphoma	10 (0.46)
C88	Malignant immunoproliferative diseases	9 (0.41)
C91	Lymphoid leukemia	505 (23.06)
C92	Myeloid leukemia	187 (8.54)
C94	Other leukemias of specified cell type	10 (0.46)
C95	Leukemia of unspecified cell type	2 (0.09)
C96	Other and unspecified malignant neoplasms of lymphoid, hematopoietic and related tissue	49 (2.24)

**Table 2 genes-15-01141-t002:** The prevalence of common and rare neoplasms in children, categorized by age, and defined by the ICD-10 classification (as described in Table 1), is presented with both numbers and percentages in brackets. The percentages of specific cancer incidence by age and legend color are shown in the columns.

Age [y]	C47 *N* = 177	C49 *N* = 161	C64 *N* = 130	C70–72 *N* = 382	C81 *N* = 144	C83 *N* = 124	C91 *N* = 505	C92 *N* = 187	Other *N* = 380	Total *N* = 2190
0 to <1	64 (36.2)	31 (19.3)	17 (13.1)	34 (8.9)	1 (0.7)	3 (2.4)	13 (2.6)	9 (4.8)	42 (11.1)	214 (9.8)
1 to <2	32 (18.1)	16 (9.9)	14 (10.8)	29 (7.6)	0 (0.0)	5 (4.0)	30 (5.9)	18 (9.6)	39 (10.3)	183 (8.4)
2 to <3	20 (11.3)	17 (10.6)	15 (11.5)	34 (8.9)	0 (0.0)	5 (4.0)	60 (11.9)	12 (6.4)	19 (5.0)	182 (8.3)
3 to <4	11 (6.2)	13 (8.1)	23 (17.7)	30 (7.9)	3 (2.1)	4 (3.2)	55 (10.9)	9 (4.8)	11 (2.9)	159 (7.3)
4 to <5	10 (5.6)	12 (7.5)	22 (16.9)	26 (6.8)	2 (1.4)	5 (4.0)	59 (11.7)	9 (4.8)	16 (4.2)	161 (7.4)
5 to <6	10 (5.6)	4 (2.5)	13 (10.0)	21 (5.5)	0 (0.0)	7 (5.6)	42 (8.3)	7 (3.7)	10 (2.6)	114 (5.2)
6 to <7	7 (4.0)	6 (3.7)	3 (2.3)	23 (6.0)	5 (3.5)	11 (8.9)	31 (6.1)	5 (2.7)	13 (3.4)	104 (4.7)
7 to <8	4 (2.3)	1 (0.6)	5 (3.8)	23 (6.0)	2 (1.4)	8 (6.5)	29 (5.7)	10 (5.3)	12 (3.2)	94 (4.3)
8 to <9	2 (1.1)	5 (3.1)	4 (3.1)	19 (5.0)	4 (2.8)	7 (5.6)	26 (5.1)	5 (2.7)	10 (2.6)	82 (3.7)
9 to <10	2 (1.1)	3 (1.9)	2 (1.5)	13 (3.4)	5 (3.5)	6 (4.8)	23 (4.6)	3 (1.6)	12 (3.2)	69 (3.2)
10 to <11	3 (1.7)	4 (2.5)	2 (1.5)	17 (4.5)	7 (4.9)	9 (7.3)	24 (4.8)	13 (7.0)	15 (3.9)	94 (4.3)
11 to <12	3 (1.7)	6 (3.7)	0 (0.0)	15 (3.9)	11 (7.6)	5 (4.0)	12 (2.4)	16 (8.6)	18 (4.7)	86 (3.9)
12 to <13	2 (1.1)	4 (2.5)	1 (0.8)	15 (3.9)	12 (8.3)	6 (4.8)	19 (3.8)	10 (5.3)	21 (5.5)	90 (4.1)
13 to <14	0 (0.0)	9 (5.6)	0 (0.0)	12 (3.1)	11 (7.6)	12 (9.7)	23 (4.6)	14 (7.5)	15 (3.9)	96 (4.4)
14 to <15	2 (1.1)	5 (3.1)	1 (0.8)	13 (3.4)	10 (6.9)	7 (5.6)	14 (2.8)	13 (7.0)	27 (7.1)	92 (4.2)
15 to <16	1 (0.6)	8 (5.0)	1 (0.8)	19 (5.0)	26 (18.1)	13 (10.5)	18 (3.6)	10 (5.3)	27 (7.1)	123 (5.6)
16 to <17	3 (1.7)	6 (3.7)	6 (4.6)	17 (4.5)	21 (14.6)	6 (4.8)	11 (2.2)	13 (7.0)	38 (10.0)	121 (5.5)
17 to <18	1 (0.6)	11 (6.8)	1 (0.8)	22 (5.8)	24 (16.7)	5 (4.0)	16 (3.2)	11 (5.9)	35 (9.2)	126 (5.8)
Legend	<5.0%	5.0–9.9%	10.0–14.9%	15.0–19.9%	≥20.0%					

**Table 3 genes-15-01141-t003:** The incidence of neoplasms in children based on age at diagnosis and the presence of cancer predisposition syndromes (CPSs).

Age, Year (s)/Month (s) Old [y/m]	CPS Presence		Statistical Analysis		
	No *N* = 1997 *n* (%)	Yes *N* = 193 *n* (%)	OR	95% CI	*p*
0 to <1 y	165 (8.3)	49 (25.4)	3.78	2.63–5.42	<0.0001
0 to <1 m	33 (1.7)	11 (5.7)	3.60	1.79–7.24	0.0001
1 to <2 m	15 (0.8)	6 (3.1)	4.24	1.63–11.1	0.0048
2 to <3 m	14 (0.7)	3 (1.6)	2.24	0.64–7.85	0.1839
3 to <4 m	13 (0.7)	3 (1.6)	2.41	0.68–8.53	0.1613
4 to <5 m	7 (0.4)	1 (0.5)	1.48	0.18–12.1	0.5225
5 to <6 m	14 (0.7)	5 (2.6)	3.77	1.34–10.6	0.0216
6 to <7 m	16 (0.8)	4 (2.1)	2.62	0.87–7.92	0.0926
7 to <8 m	16 (0.8)	3 (1.6)	1.95	0.56–6.77	0.2313
8 to <9 m	13 (0.7)	3 (1.6)	2.41	0.68–8.53	0.1613
9 to <10 m	7 (0.4)	5 (2.6)	7.56	2.38–24.1	0.0004
10 to <11 m	8 (0.4)	2 (1.0)	2.60	0.55–12.4	0.4891
11 to <12 m	9 (0.5)	3 (1.6)	3.49	0.94–13.0	0.1408
1 to <2 y	171 (8.6)	12 (6.2)	0.71	0.39–1.30	0.2609
2 to <3 y	157 (7.9)	25 (13.0)	1.74	1.11–2.74	0.0144
3 to <4 y	134 (6.7)	25 (13.0)	2.07	1.31–3.26	0.0014
4 to <5 y	147 (7.4)	14 (7.3)	1.04	0.59–1.85	0.8848
5 to <6 y	106 (5.3)	8 (4.1)	0.77	0.37–1.61	0.5997
6 to <7 y	100 (5.0)	4 (2.1)	0.77	0.37–1.61	0.5997
7 to <8 y	89 (4.5)	5 (2.6)	0.57	0.23–1.42	0.3005
8 to <9 y	80 (4.0)	2 (1.0)	0.25	0.06–1.03	0.0606
9 to <10 y	62 (3.1)	7 (3.6)	1.17	0.53–2.60	0.8565
10 to <11 y	85 (4.3)	9 (4.7)	1.10	0.55–2.23	0.9334
11 to <12 y	81 (4.1)	5 (2.6)	0.64	0.26–1.60	0.4472
12 to <13 y	83 (4.2)	7 (3.6)	0.87	0.40–1.90	0.8698
13 to <14 y	93 (4.7)	3 (1.6)	0.32	0.10–1.03	0.0678
14 to <15 y	90 (4.5)	2 (1.0)	0.22	0.05–0.91	0.0351
15 to <16 y	120 (6.0)	3 (1.6)	0.25	0.08–0.78	0.0163
16 to <17 y	114 (5.7)	7 (3.6)	0.62	0.29–1.35	0.2966
17 to <18 y	120 (6.0)	6 (3.1)	0.51	0.22–1.17	0.1420

**Table 4 genes-15-01141-t004:** Age ranges relevant to the incidence of CPS-associated neoplasm in children.

Age, Year (s)/Month (s) Old [y/m]	CPS Presence		Statistical Analysis		
	No *N* = 1997 *n* (%)	Yes *N* = 193 *n* (%)	OR	95% CI	*p*
11 to <18 y	701 (35.1)	33 (17.1)	1.00	reference	
1 to <11 y	1131 (56.6)	111 (57.5)	2.08	1.40–3.11	0.0002
2 to <12 m	117 (5.9)	32 (16.6)	5.81	3.44–9.81	<0.0001
0 to <2 m	48 (2.4)	17 (8.8)	7.52	3.91–14.5	<0.0001

## Data Availability

All data are presented in the manuscript and Appendix A.

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
