# Peer review of "The Prevalence of Cancer Predisposition Syndromes (CPSs) in Children with a Neoplasm: A Cohort Study in a Central and Eastern European Population"

_genes, 2024, doi:10.3390/genes15091141_

Round 1
Reviewer 1 Report
Comments and Suggestions for Authors
This paper is interesting, but some points need to be revised:
- Abstract should be structured. Revise it.
- Lines 60-65. It is not clear what is the purpose of this paper. In addition this sentence should be moved to Methods. "The data were collected between 2000 and 2019 in 64 Karol Jonscher’s Clinical Hospital of Poznan University of Medical Sciences. "
- Lines 166-170: "To summarize, two peaks of cancer incidence.. adolescents (16.9%)." How do the authors can explain this? Improve this.
- Lines 288-290: "In our cohort, conducting diagnostic tests for CPSs was considered in case of rare and unspecific cancer occurrence, phenotypic features, recurrence or multiplied neoplasm detection, and burdened family history" This part is not clear. Improve genetic issue.
- Lines 302-306: Improve conclusion. What does this paper add new to the literature?
Comments on the Quality of English LanguageMinor editing of English language required
Author Response
Dear Reviewer,
We are very grateful for the review of the article ”Prevalence of cancer predisposition syndromes (CPSs) in children with a neoplasm - a cohort study in a Central and Eastern European population”. We would like to address your comments and suggestions - please see the attachment.

Reviewer 2 Report
Comments and Suggestions for Authors
The manuscript "Prevalence of cancer predisposition syndromes (CPSs) in children with a neoplasm - a cohort study in a Central and Eastern European population" is well-organized but contains several lengthy and complex sentences.
General Considerations:
- Simplifying these sentences can improve readability, check English grammar and the presence of typo
Abstract
-Line 18: "The causes of pediatric cancers often remain unknown..." Consider rephrasing for clarity: "The etiology of pediatric cancers is often unclear; however, advancements in genetics have identified significant roles for genetic disorders in their development."
Introduction
-Line 37: "Cancers in children remain rare diseases..." - This phrase could be more precise, e.g., "Pediatric cancers are relatively rare, with an estimated annual incidence of up to 400,000 cases globally."
-Line 40: Clarify "Knudson made a two-hit hypothesis" with "Knudson proposed the two-hit hypothesis, which involves the inactivation of both alleles of a tumor suppressor gene leading to cancer development."
-In addition in my opinion the Introduction is too brief, I suggest to add more background, and to add the limitations of previous studies that can help us understand why it was important to carry out such a study to better understand pathogenetic mechanisms that are not yet clear.
Discussion
This section of the study, like the Introduction, is also addressed very superficially and briefly.
-The discussion should connect the findings more clearly to the broader field, discussing how they compare to existing literature and what new insights they provide.
-Highlight any limitations in the study and suggest directions for future research.
Comments on the Quality of English LanguageModerate editing of English language required
Author Response
Dear Reviewer,
We are very grateful for the review of the article entitled ”Prevalence of cancer predisposition syndromes (CPSs) in children with a neoplasm - a cohort study in a Central and Eastern European population”. We would like to address your comments and suggestions - please see the attachment.

Reviewer 3 Report
Comments and Suggestions for Authors
Authors present a central and eastern european population study on prevalence of cancer predisposition syndromes (CPSs) in children with neoplasm. 193 children with CPS treated at single center were included, which makes 8,8% of all cancer pediatric population. NF-1, Li Fraumeni and Down syndrome were the most common one. This is a nicely formed study which sheds new light on the epidemiology of CPS. However, authors fail to make any clinicaly relevant statements to their research, i.e. the clinical consequence of this study needs to be explained. Discussion is way to short - I suggest to include a literature review on CPS and previous studies and compare results.
Comments on the Quality of English LanguageModerate changes.
Author Response

(The authors gave the same response as above.)

Round 2
Reviewer 1 Report
Comments and Suggestions for Authors
Good
Reviewer 3 Report
Comments and Suggestions for Authors
Sufficient response to reviewer remarks.